# Hierarchical Planning in the IPC

**G. Behnke**[*], **D. Höller**[*], **P. Bercher**[*], **S. Biundo**[*], **D. Pellier**[†], **H. Fiorino**[†], and **R. Alford**[‡]

[*]Institute of Artificial Intelligence, Ulm University, 89081 Ulm, Germany
{gregor.behnke, daniel.hoeller, pascal.bercher, susanne.biundo}@uni-ulm.de
[†]University Grenoble Alpes, LIG, F-38000 Grenoble, France
{damien.pellier, humbert.fiorino}@imag.fr
[‡]The MITRE Corporation, McLean, Virginia, USA
ralford@mitre.org

## Abstract

Over the last years, the amount of research in hierarchical planning has increased, leading to significant improvements in the performance of planners. However, the research is diverging and planners are somewhat hard to compare against each other. This is mostly caused by the fact that there is no standard set of benchmark domains, nor even a common description language for hierarchical planning problems. As a consequence, the available planners support a widely varying set of features and (almost) none of them can solve (or even parse) any problem developed for another planner. With this paper, we propose to create a new track for the IPC in which hierarchical planners will compete. This competition will result in a standardised description language, broader support for core features of that language among planners, a set of benchmark problems, a means to fairly and objectively compare HTN planners, and for new challenges for planners.

## Introduction

When the International Planning Competition (IPC) started out in 1998 it aimed to include both classical and hierarchical planners as competitors (McDermott 2000). To provide a fair competition and foster further research in planning, a unified description language for planning problems, the Planning Domain Definition Language (PDDL), was created. The main objective of PDDL was to separate the description of the physics of a domain from additional advice. Planners should operate on the physics of the domains only. This way the competition shows how good the underlying domain-independent techniques are for solving planning problems based on the description of a domain's physics. Notably, PDDL also included features for expressing hierarchical planning problems. Hierarchical planning was not included in the first IPC, as there were no competitors.

The second and third IPC subsequently included a separate track for "Hand-Tailored Systems", where the planner was allowed to have additional advice for each individual domain. This was within the spirit of hierarchical planning at the time (see e.g. SHOP's participation (Nau et al. 1999)). Hierarchical structures in a planning problem were – at the time – almost exclusively seen as additional advice given to the planner in order to speed up planning. The hierarchy was viewed as specifying pre-defined recipes for a given objective. After the third IPC, no further attempt was made to include hierarchical planning into the IPC.

Since the third IPC in 2002, the research in hierarchical planning has progressed significantly. Especially the view what hierarchical planning is, has changed sharply. A hierarchy over a planning problem is not (only) a means to provide advice to the planner, but a general means to specify parts of the domain. Just like non-hierarchical structures, it may be used to model advice (as done before), but also to describe additional physics of the domain. Research in hierarchical planning has, over the last decade, mostly focussed on a single formalism: Hierarchical Task Network (HTN) Planning (Erol, Hendler, and Nau 1996)[1]. The physics expressible in HTN planning cannot be equivalently expressed in classical planning (Höller et al. 2014; Höller et al. 2016). In fact, HTN planning allows for expressing undecidable problems, like Post's Correspondence Problem or Context-free Grammar Intersection (Erol, Hendler, and Nau 1996). Based on the researcher's focus on a single, theoretically understood formalism, we think the time is right for an IPC track in which HTN planners compete. Creating such a competition was one of the motivations behind organising the Workshops on Hierarchical Planning at ICAPS 2018 and 2019. A discussion on the details of a (potential) IPC track for HTN planning is planned for this year's workshop.

The HTN planning community lacks a strong unifying force like the IPC is for classical planning. Due to the lack of a competition in HTN planning, there is no precisely agreed-upon semantics for modelled planning problems[2], no unified input language readable by all planners, and no standardised benchmark to compare planners. A competition should resolve these problems and provide a solid basis for future work, as was the case for classical planning. We therefore propose to add a new track to the IPC in which HTN planners will compete. In this paper, we first give a brief overview of HTN planning, report on results of an online questionnaire on an HTN IPC track, and then present the details of our prosed HTN IPC track.

---

[1]Note that there is also research within hierarchical planning that does not qualify as HTN planning, such as work on HGN and GTN planning (Shivashankar et al. 2013; Alford et al. 2016b). See Bercher, Alford, and Höller (2019) for an overview.

[2]Although there is a theoretical formalism almost all planners add their individual bells and whistles making them incomparable.

# HTN Planning – State of the Art

We start by giving a quick overview over HTN planning, by reporting on theoretical and practical results related to it.

## Theory

While classical planning has only one type of tasks – actions – HTN planning distinguishes two types: primitive tasks (also called actions) and abstract tasks. Only actions hold preconditions and effects, which are defined in the usual (STRIPS or ADL) way. In contrast to actions, abstract tasks cannot be executed directly. Instead, each abstract task $A$ has a set of associated decomposition methods $(A, tn)$ that allow for achieving $A$ by performing the tasks (that may, again, be primitive or abstract) contained in $tn$. Here $tn$ is a task network, which gives HTN planning its name. A task network is a partially ordered multi-set of tasks.

The objective in HTN planning is given by an initial state and an initial abstract task (replacing the state-based goal definition in classical planning[3]). A solution is any sequence of actions that is executable in the initial state, provided it can be obtained from the initial abstract task by repeatedly applying decomposition methods. In this structure, HTN planning is very similar to formal languages, where terminals correspond to actions and non-terminals to abstract tasks (Höller et al. 2014; Barták and Maillard 2017).

A simplistic formal description of HTN planning was proposed by Geier and Bercher (2011). An equivalent formulation in terms of combinatorial grammars was given by Barták, Maillard, and Cardoso (2018). The simplistic formalism by Geier and Bercher is theoretically well understood, witnesses by the multitude of papers based on it.

The plan existence problem in HTN planning in its general form is undecidable (Erol, Hendler, and Nau 1996). There are, however, several restrictions to the formalism that make it decidable. The most notable one is totally-ordered HTN planning. Here, all task networks in decomposition methods must be sequences of tasks instead of partially-ordered sets. In contrast to general HTN planning, ground, totally-ordered HTN planning is EXPTIME-complete (Erol, Hendler, and Nau 1996).

## Planners

In the past, a multitude of HTN planners have been developed. This includes the older systems which were mostly based on uninformed search, like SHOP (Nau et al. 1999), SHOP2 (Nau et al. 2003), or UMCP (Erol, Hendler, and Nau 1994). Recently, the portfolio of available methods to solve HTN planning problems has increased significantly. Many ideas developed for classical planning have been transferred to HTN planning and proved successful there. This lead to a wide range of (partially) available HTN planners.

- FAPE (Dvorak et al. 2014), a temporal HTN planner with strong pruning techniques
- PANDA (Bercher et al. 2017), a plan-space planner using heuristic search

---

[3]Note that every classical planning problem can be converted into an equivalent HTN planning problem (Erol, Hendler, and Nau 1996).

- PANDApro (Höller et al. 2018; Höller et al. 2019b), a progression-based planning system using heuristic search
- GTOHP (Ramoul et al. 2017), a planner based on intelligent grounding and blind search
- HTN2ASP (Dix, Kuter, and Nau 2003), a planner that translates (totally-ordered) HTN planning problems into answer set programming
- HTN2STRIPS (Alford et al. 2016a), a planner translating HTN planning problems into a sequence of classical planning problems
- totSAT (Behnke, Höller, and Biundo 2018) and Tree-Rex (Schreiber et al. 2019), planners that translates (totally-ordered) HTN planning problems into propositional logic
- partSAT (Behnke, Höller, and Biundo 2019a; 2019b), a planner based on a translation into propositional logic

It is noteworthy that four of the planners (GTOHP, totSAT, Tree-Rex, and HTN2ASP) are specifically designed for solving totally-ordered HTN planning problems. Further, HTN2STRIPS is based on an older encoding of totally-ordered HTN planning problems into classical planning (Alford, Kuter, and Nau 2009).

# Online Questionnaire

In order to better understand the needs and interest of the community when organising a deterministic HTN IPC track in 2020, we conducted a short online questionnaire and sent the link for participation to the planning community.

We obtained 23 responses from 20 different research groups. 80% of the respondents would like to participate in an HTN IPC track in 2020. Those not wishing to participate often justified their refusal with the lack of HTN standards (language, benchmarks, tools). We would argue that starting the competition is the only way to produce such a standard in the first place. As such, organising an HTN IPC competition seems to respond to the needs of the community.

The main motivation for those wishing to participate is to be able to objectively compare the performance of HTN planners. However, this was not the only reason given by respondents. People also wished to increase the expressiveness of the PDDL language by adding HTN language standards.

We also asked what the respondents thought to be absolute necessities for the competition. The four top answers ranked in order of importance are:

1. a standard language definition for HTN planning,
2. an accessible HTN benchmarks repository,
3. an HTN plan validator, and
4. an HTN parser.

In terms of tracks, most people think that optimal and satisfying should be enough for a first IPC competition. Some people also expressed interest in participating in a temporal track. For the first HTN IPC, we propose to restrict the competition to a non-temporal setting, with the option of a temporal track to be added in the future. We think that the competition should focus on the core of HTN planning and thus should compare the planners on the absolute essentials.

In terms of metrics that should be used to compare HTN planners, opinions are divided. 50% of the respondents think

that the IPC score is suitable for an HTN IPC track. 50% think that we need to devise new metrics, which they could propose as free text answers. Respondents proposed to use (1) the number of backtracking operations performed by the planner, (2) the depth the explored search space, (3) the size of the method library used by the planner, and (4) the expressivity and explicability of the solution.

We propose to keep the IPC score as opposed to these proposals. First, we argue that all planners in the competition *must* use the same HTN method library to ensure that the results of the competition are correct, objective, and fair. If every planner would be allowed its own set of methods, we would not measure the performance of the planner, but the ingenuity of the modeller. Furthermore, the set of solutions the problems encodes would differ from planner to planner – which makes little to no sense, as a comparison between the mechanics of planners is not possible anymore.

Using either the number of backtracking operations or the depth of the explored search space would restrict the competition to search-based planners, as e.g. SAT-based planners can't produce these measures. We think that these two metrics stem from the idea that the planners may use their own HTNs for the domains, as both the number of backtracking operations as well as the depth of the search space are metrics to measure the "difficulty" of an HTN domain. Further, to measure these metrics, we would have to rely on the information reported by the planners about their internal search, which is somewhat strange for an objective competition. Similarly, measuring the explicability of solutions objectively is hard or even impossible at the moment.

In terms of HTN planning language, a majority of people thinks it is better to add hierarchical planning concepts to the PDDL language rather than redefining a new language from scratch. We elaborate on this in the next section.

Concerning benchmarks, most respondents wish to participate to the definition of HTN benchmarks. We asked respondents which types of domains they wished to see as part of the benchmark set (see Fig. 1). Note that these answers have to be considered with care, as most of the mentioned types of domains are also domains used in the classical IPC. HTN planning is *not* a means to solve classical planning problems faster, but a complex combinatorial problem in its own right. There are structural restrictions (e.g. PCP and Grammar Intersection domains) to plans that cannot be expressed by preconditions and effects of actions, while it is possible to formulate them in an HTN domain (Erol, Hendler, and Nau 1996; Geier and Bercher 2011; Höller et al. 2014). Further, the domain's hierarchy can express physics that is not modelled in the primitive actions and is thus an integral part of the problem definition. For example, in a transport domain, the location of each truck can be fully modelled in an HTN problem without introducing state variables pertaining to the location of the truck. Without the hierarchy, the primitive action theory in this model makes no sense at all. As such, the mentioned domain proposals should be viewed as potential topics for modelling domains with highly complex restrictions on plans and not as the suggestion that the domain should express the same mechanics as its classical counterpart.

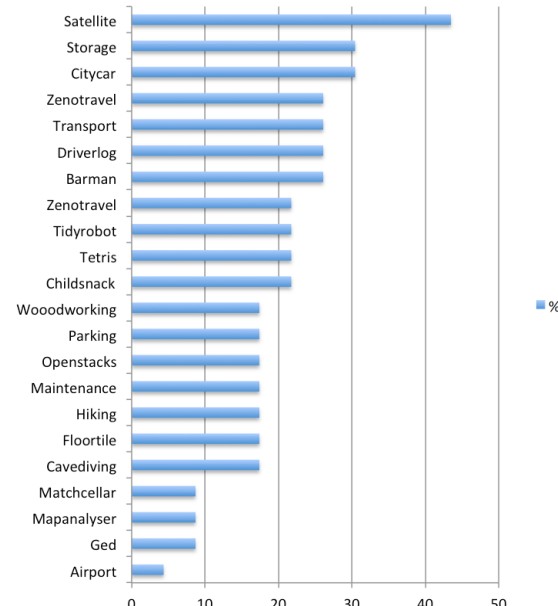

Figure 1: Types of domains proposed by respondents for HTN IPC track

## Common Description Language

Organising the first IPC track on HTN planning also involves defining a fixed input language for all participating planners. For classical planning, there is one universally accepted description language – PDDL. The situation for HTN planning is drastically different. Of the planners mentioned in the previous section, PANDA, PANDApro, totSAT, and partSAT accept the same input language. Similarly GTOHP and Tree-Rex have a common input language, which is distinct from PANDA's. All other planners use their own individual description languages. Both groups of planners using the same input language also share a common parser and grounder. The two formats are somewhat incompatible and the format of GTOHP and Tree-Rex does only support totally-ordered HTN planning problems.

Some of the description languages are based on PDDL, while others, like SHOP's language and ANML (Smith, Frank, and Cushing 2008) are drastically different. We think that a language based on PDDL is the most sensible way to define the HTN description language, as it will allow for close contact between the HTN and classical planning communities (tools for preprocessing like Fast Downward's grounder which transforms the planning problem into the SAS+ format (Helmert 2009) might, e.g. be used in both communities). A proposal for an HTN description language (Höller et al. 2019a) has been accepted at this year's Workshop on Hierarchical Planning at ICAPS.

The proposed language is based on the STRIPS part (language level 1) of the PDDL 2.1 language definition (Fox and Long 2003). In this paper, we want to introduce the extensions made and explain some of the design decisions. In the original paper (Höller et al. 2019a), we give a much broader discussion, especially with respect to several alternative def-

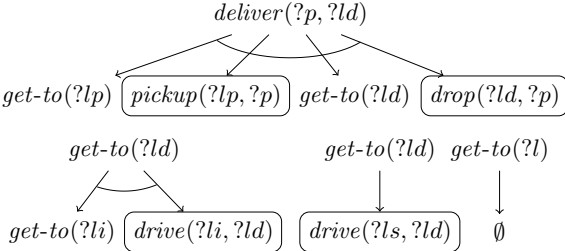

Figure 2: The method set of a simple transport domain. Actions are given as boxed nodes, abstract tasks are unboxed. All methods are totally ordered (source of figure: Höller et al., 2019a).

initions from related work and also provide a full EBNF definition of the entire language.

We want to introduce the language using a simplistic transport domain that is illustrated in Figure 2. There is only a single transporter that has to deliver packages. The method set is designed to exemplify the features of the language (so we know that there are more compact equivalent definitions). The overall deliver task can be decomposed (by the method given at the top) into a *get-to* task that moves the transporter to the package, a *pick-up* action, another *get-to* task that makes the transporter move to the position the package shall be delivered, and a *drop* action. For the *get-to* task, there are three methods (given at the bottom, from left to right): the first one is recursive and makes the transporter get to an intermediate position from which the target position can be reached directly by a primitive *drive* action, the second one can be used when it is already directly reachable by a *drive* action, and the third one can be used when the transporter is already at its destination, it decomposes the *get-to* into an empty task network. The search is started with one or more *deliver* tasks.

When we look at the proposed input language, the beginning of the domain definition is equal to PDDL, defining the domain name, a type hierarchy, and predicates.

```
1 (define (domain transport)
2   (:types location package - object)
3   (:predicates
4       (road ?l1 ?l2 - location)
5       ...)
```

The first element we had to add is the definition of *tasks*. There are two common ways to define them: as an explicit enumeration, or, by simply *using* the tasks in the method specifications and defining the set of abstract tasks as the union of all tasks used in methods. We decided to require an explicit definition of abstract tasks and tried to keep it close to the one of actions in PDDL (in fact, both actions and abstract tasks are quite similar in HTN planning) due to the following reasons:

- In our opinion, an implicit definition is against the design spirit of PDDL. Take the predicate definitions as an example. It would be sufficient to omit the explicit definition of predicates and just use them in action definitions. PDDL opted for the first variant, an explicit definition.

- There are hierarchical planning approaches where abstract tasks hold preconditions and effects. A definition as given here allows for a simple extension to support such approaches.

- Lastly, there are benefits to being able to explicitly model the types of parameters of abstract tasks (see how tasks are used in method definitions).

The following listing defines the two abstract tasks used in the transport example.

```
6   (:task deliver :parameters (?p - package
        ?l - location))
7   (:task get-to :parameters (?l - location))
```

There is a single method decomposing the *deliver* task:

```
8   (:method m-deliver
9     :parameters (?p - package
          ?lp ?ld - location)
10    :task (deliver ?p ?ld)
11    :ordered-subtasks (and
12      (get-to ?lp)
13      (pick-up ?ld ?p)
14      (get-to ?ld)
15      (drop ?ld ?p)))
```

The method definition starts with its name (here: `m-deliver`) and is followed by the parameter definition.

The parameters could, again, be either defined explicitly (like we did), or implicitly by just using them. Having an explicit definition has two advantages:

- Having an explicit definition allows for consistency checks by the planning system.

- By having the method parameters specified explicitly, one can restrict the applicability of a method via the type definition. As an example, consider a transport domain with several types of packages. Some of them might need special ways to be delivered, e.g. hazardous materials.

In the proposed language, all variables used in a method must be declared as parameters of that method – similar to the parameters of PDDL actions. This includes variables used for the specification of the abstract task that is decomposed, in the specification of the subtasks, and the constraint set.

Next, the task that the method decomposes is specified (line 10). It has to be defined in the domain as given before.

The method definition closes with the subtasks (starting in line 11). In the given method, the tasks shall be totally ordered. To enable a compact domain definition, the modeller can use the `:ordered-subtasks` keyword to indicate that all subtasks shall be totally ordered in the order in which they are specified. As subtasks, any action or task defined in the domain can be used, while their arguments can be constants and any of the parameters of that method.

In the next method, we see how to define the order of subtasks explicitly.

```
16    (:method m-drive-to-via
17      :parameters (?li ?ld - location)
18      :task (get-to  ?ld)
19      :subtasks (and
```

```
20        (t1 (get-to ?li))
21        (t2 (drive ?li ?ld)))
22     :ordering (and
23        (t1 < t2)))
```

The subtasks are "marked" with IDs (here: `t1` and `t2`). The order is specified using these IDs. Supporting these two ways to specify ordering, we enable a compact definition of totally ordered networks (using the first variant) without loosing the expressivity to specify arbitrary partial order.

Many HTN planners allow for method preconditions (see Höller et al. (2019a) for a discussion of this feature). We decided to include them as well.

```
24    (:method m-already-there
25       :parameters (?l - location)
26       :task (get-to ?l)
27       :precondition (tAt ?l)
28       :subtasks ())
```

Here, the *get-to* task may be achieved by doing nothing when the transporter is already at its final position. This is checked by the state-based precondition (line 27).

HTN planners from the literature support various forms of conditions, like e.g. equality and inequality of variables, constraints on the types, but also conditions on the state that need to hold between two tasks. Though we decided to keep the initial formalism simple, we added a `constraints` section to our definition that can be seen in the following listing (line 32).

```
29    (:method m-direct
30       :parameters (?ls ?ld - location)
31       :task (get-to ?ld)
32       :constraints
33          (not (= ?li ?ld))
34       :subtasks (drive ?ls ?ld))
```

In the first language version, only equality and inequality are supported, but the presence of the constraints section allows for an easy extension in future language versions.

The action definitions remain as defined in PDDL2.1.

```
35    (:action drive
36       :parameters (?l1 ?l2 - location)
37       :precondition (and
38          (tAt ?l1)
39          (road ?l1 ?l2))
40       :effect (and
41          (not (tAt ?l1))
42          (tAt ?l2)))
43    ...)
```

Another important change is made to the problem definition. Here, the initial task network is specified (starting in line 6), its definition is similar to the definition of method sub-networks (therefore we included the empty `:ordering` and `:constraints` in the next listing to show the similarity, these might, of course, be omitted).

```
1 (define (problem p)
2  (:domain transport)
3  (:objects
4   city-loc-0 city-loc-1 city-loc-2 -
       location
```

```
5   package-0 package-1 - package)
6  (:htn
7   :tasks (and
8     (deliver package-0 city-loc-0)
9     (deliver package-1 city-loc-2))
10   :ordering ()
11   :constraints ())
12  (:init
13   (road city-loc-0 city-loc-1)
14   (road city-loc-1 city-loc-0)
15   (road city-loc-1 city-loc-2)
16   (road city-loc-2 city-loc-1)
17   (at package-0 city-loc-1)
18   (at package-1 city-loc-1)))
```

Another point we want to highlight is the specification of the problem class given in line 6. It is specified by the keyword `:htn`. There are many slightly different problem classes in hierarchical planning (see Bercher, Alford, and Höller (2019) for an overview). Some allow, e.g., for task insertion. The explicit definition in the problem file allows to extend the language standard to other classes than common HTN planning.

## Tracks

As in the IPC for deterministic, classical planning, we propose to run competitions for optimal, satisficing, and agile planning. Further we propose to split the competition between the types of problems handled by the planners. Since there is a large group of existing HTN planners that is restricted to totally-ordered HTN planning problems, we propose to include a separate track for those planners in which all input problems are totally-ordered. Naturally, all HTN planners that are capable of handling general HTN planning problems can take part in a totally-ordered competition, but should be at a disadvantage. In a second set of sub-tracks, the planners that are able to solve general, i.e. partially-ordered, HTN planning problems will compete. As a result, we propose six tracks: optimal totally-ordered, satisficing totally-ordered, agile totally-ordered, optimal general, satisficing general, and agile general.

## Timeline and Organisation

We propose to roughly follow the usual schedule of the IPC.

| | |
|---|---|
| **May – July 2019** | Agreeing on a common input language for all planners. |
| **July 2019** | Announcement of the track Call for domains Call for expression of interest |
| **October 2019** | Registration deadline |
| **November 2019** | Demo problems provided |
| **January 2020** | Submission of preliminary planner versions |
| **February 2020** | Domain submission deadline |
| **April 2020** | Final planner submission deadline |
| **May 2020** | Paper submission deadline |
| **May 2020** | Contest run |
| **June 2020** | Presentation of the results at ICAPS 2020 |

As did the first IPC track on unsolvability (Muise and Lipovetzky 2015), we also expect that there will be many technical issues with the submitted planners. We assume that most of them will be centred around correct support of the input language, for which issues will usually take some time to debug and fix. Further, HTN planners have to provide the decompositions they used as their output as well – in order to be able to verify their solutions in time. Without these decomposition, verification is $\mathbb{NP}$-complete (Behnke, Höller, and Biundo 2015) and thus may take a long time. We will use the preliminary submissions of the planners to validate that their outputs are correctly formatted and can be verified against the domain. As such, we propose a first submission of the planners early on, so that we can test them sufficiently before the actual competition.

## Scoring and Setting

For scoring the planner, we will adopt the metrics used in the last deterministic, classical IPC. We further propose to use the same technical setting (1 core, 8 GB of RAM, and 30 minutes). All planners will be provided with the same HTN planning domain, i.e. set of primitive actions, abstract tasks, and decomposition methods, the same planning problem, i.e. initial state and initial abstract task, and are not allowed to have any additional domain-dependent information. They have to output both the solution, i.e. a sequence of primitive actions, as well as a witness that this solution is obtainable via decomposition form the initial abstract task.

## Domains

As noted before, there is no large set of available benchmarking domains for HTN planners, which would have to be created for the competition. The IPC has so far relied for the domains in the competition on an open call for the community to submit domains. We propose will also call for such community-provided domains (once a input language has been fixed). In addition, we propose to add a new mode of providing benchmark domains that has been recently adopted by the SAT community: Bring Your Own Benchmark (BYOB). The SAT Competition 2017[4] and 2018[5] used BYOB. In it, each competing program – in our case planner – is required to submit a domain with 20 instances. Of these 20 instances, the planner of the submitter must be able to solve at *most* 10. This requirement forces submissions of domains that are not advantageous to the planner of the submitter. This encourages the submission of problems that are of medium difficulty for the individual planner. We argue that they will both provide a good basis for comparison against other planners as well as a good starting point for future scientific investigations based on the difficulties in them.

From a content perspective, we especially encourage submissions of domains that pose problems which cannot be expressed in classical planning, i.e. in PDDL[6]. Such domains exists, like PCP and Grammar Intersection, but it would be

interesting to see where HTN planning can further use its high expressive power.

## Discussion

With a new IPC track on HTN planning, we think that the research in hierarchical planning will be more focussed and successful in the future. As a result of the competition, we expect that

- the HTN community will agree upon a set of core features supported by every planner and an input language for it, which is important for users of HTN planning, as well as for comparing systems against each other.

- a set of benchmark domains will be available, allowing for better judging and fairer comparisons of planners than is currently possible.

- a core of (relatively) error-free software to be used by many planners will emerge, like Fast Downward (Helmert 2006) for classical planning, allowing for both easy use by planning researchers and users of planning technology.

- hints for future research in HTN planning will be given.

While the first three points are valuable to the community and outsiders wanting to use HTN planning, we want to emphasise the fourth. The results of planning competitions will regularly show the weaknesses of the so-far developed approaches and techniques. These weaknesses are valuable information and point out where planners can and should be improved in the future.

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
