# OpenReview forum: "Hierarchical Planning in the IPC"
_icaps-conference.org/ICAPS/2019/Workshop/WIPC_

### Official Review · AnonReviewer3 · 2019-04-23
**HTN tracks are valuable addition to IPC**

**Rating:** 9
**Confidence:** 4

**Review:**

The paper proposes to establish an hierarchical planning track in
addition to the classical, probabilistic and temporal IPC tracks. It
analyzes the current state of the art in hierarchical planning, presents
initial results from a questionnaire on a possible competition and
provides initial ideas on the form of the tracks and how the timeline,
scoring method and benchmarks will look like. Most of this is based on
experience from the classical planning tracks (which is a good thing),
and the one notable change (the community provided BYOP domains) are
something I appreciate and would like to see in action.

The topic is a perfect fit for the workshop and will certainly be
interesting to everyone interested in hierarchical planning. What I find
particularly important (as someone not from the field) is the fact that
of the mentioned 9 hierarchical planners that have been released less
than 15 years ago, only 1 (FAPE) does not share a developer with the set
of the paper's authors, indicating that the organizers of the HTN track
aim for a set of rules and an input language that reflects the interest
of as many community members as possible.

From my point of view, all decisions are well justified. The one thing I
would like to see added to the paper is the initial proposal for the
hierarchical PDDL variant the authors have in mind. I am aware that the
language is presented at another ICAPS workshop, but this will most
likely be the most controversial point for the planned competition. I
believe that members of the classical planning community, who have a lot
of experience with PDDL, might be able to provide valuable input
regarding the PDDL extensio and think that this might spawn a fruitful
discussion at the workshop. I would therefore appreciate it if the
authors add a (possibly shortened) PDDL description to the paper, and
then I would propose to turn the paper into a full submission with 30
minutes discussion slot at the workshop (if time allows).


Minor comments:
p1: Over the last year -> years
p1: to compare against each other. -> to compare.
p2: transferred to HTN planning problem and -> transferred to HTN
planning and
p2: This lead to a wide range -> This led to a wide range
p2: planner that translates -> planner that translate
p3: to this year Workshop on -> to this year's Workshop on
p4: Without these decomposition -> Without this decomposition
p4: We propose will also call -> We will also call

---

### Official Review · AnonReviewer2 · 2019-04-25
**Hierarchical Planning in the IPC**

**Rating:** 7
**Confidence:** 3

**Review:**

I think this is a good paper for the workshop, and its great to see a new community rallying to get a new competition; most importantly with what appear to be available and motivated organisers: I think this initiative should definitely be encouraged.

The paper gives a good overview of the current state of the art in hierachical planning; has evidence of good support from the community via a survey (I'm very surprised there are 20 distinct research groups working in the area, this is indeed positive!); and shows well thought through considerations in the proposal.

I have used the remainder of the review to discuss potential challenges and issues, but this should not in any way be considered discouragement from holding a competition, just perhaps add some things that the organisers might want to consider.

Other Comments:

"Over the last year" Do you really mean year in singular, or 'in recent years'?

"After the third IPC, no further attempt was made to include hierarchical planning into the IPC." To play devil's advocate here for a second: was there a good reason for this: technical/lack of interest/other?  If the HTN track is brought back is there a reason to expect it would now succeed?  (I see there's some argument later here from the questionnaire, it might be worth at least noting up front that a questionnaire does suggest support).

"A hierarchy over a planning problem is not (only) a means to provide advice to the planner, but a general means to specify parts of the domain."
This is interesting, because as an outsider to the HTN community, I still have the impression that HTN planners are being 'helped' by the decomposition structure, so this is extra control knowledge; rather than that it is a harder problem (I'm aware of the complexity results, although not really wise to whehter many/any of the problems solved in practice for evaluation are actually the harder ones (just like most classical planning benchmarks are not actually even PSPACE hard) and we learn later that many planners restrict the general problem, to totally ordered HTN planning).

"Research in hierarchical planning has, over the last decade, mostly focussed on a single formalism: Hierarchical Task Network (HTN) Planning (Erol, Hendler, and Nau 1996)" I think the text in this paragraph does make for a strong argument as to why HTN planning is in a good position to have a competition.

I also fully support the arguments that having a common language for allowing empirical comparison will be very important and valuable to the community, both in terms of scientific value to ensure evaluation is easier and more sound; and in terms of directing focus to tackle certain key challenges. It might be a challenge now to encourage a community that has many existing languages to converge to use one, but I believe it's a worthy goal, and a competition might provide the motivation there.

Getting agreement on a language could also be a significant challenge, given hierachical planners are typically also particularly expressive in terms of language features handled.  One thing I've observed in the hybrid planning community is that as people are wanting to support more and more features, they are starting to diverge from taking PDDL as input, rather than using it as is, or extending it as needed: which makes empirical comparison more difficult and loses the advantages of comparability; so this may suggest challenges in getting people to agree to and stick with a language; although starting with a 'classical' and optimal track is probably a good idea here (whether people will then use this in the literature, or feel the need to use a more expressive language is worth consideration).

The paper gives a good overview of current HTN planners, and there are clearly a widely varying set of approaches.  Just a couple of questions:
"(partially) available HTN planners." what do you mean by partially available?
Also are some/all these planners domain independent, what is required to run them: just a model (and does this model contain extra control knowledge?), domain specific heuristics etc.?

The questionnaire gives a strong rebuttal to my earlier question of whether there would be interest now, it's great that there seems to be quite a bit.  Having said that, the challenges of getting a planner ready for a competition especially one for which a new language must be supported might hurt the conversion rate of interest to entries somewhat (but this is true for any competition track, and a strong initial interest certainly suggests feasibility).

The other challenge for tracks with smaller communities is getting a continued stream of organisers: people often want a competition to participate in; but the convention that organisers do not enter (for obvious reasons), and the work involved in running the competition, means it can be difficult to get people to run one. This has seen some tracks fold, e.g. preferences, and others, e.g. temporal, struggle somewhat in recent years.  So the community needs to be big and enthusiastic enough not only to get entrants, but also organisers.

Is the competition timeline planned to coincide with any other competitions/tracks or to be stand alone?

I'd imagine allowing authors to debug their planners on the competition domains will be difficult to manage fairly, but this is something that other tracks have had to consider too, so it's worth discussing what happened with previous organisers.  What is the difference between having a preliminary then final submission versus giving participants domains on which to test the planners?  Will the organisers do all the tests or the developers of the planners?  Will the planners be tested on actual competition domains, or just sample problems?  Will there be restrictions to what changes authors are allowed to make between preliminary and final submission?

I think compulsory benchmark submission is likely wise, especially since there is a lack of existing benchmarks, otherwise the burden on organisers will be high: even in planning competitions where competitors have been allowed to submit benchmarks that are favourable to their planners, uptake has been relatively low, so it might be a challenge to get enough benchmarks.